# In Silico Insights into Protein–Protein Interaction Disruptive Mutations in the PCSK9-LDLR Complex

**DOI:** 10.3390/ijms21051550

**Published:** 2020-02-25

**Authors:** William R. Martin, Felice C. Lightstone, Feixiong Cheng

**Affiliations:** 1Genomic Medicine Institute, Lerner Research Institute, Cleveland Clinic, Cleveland, OH 44195, USA; martinw3@ccf.org; 2Biosciences and Biotechnology Division, Physical and Life Sciences Directorate, Lawrence Livermore National Laboratory, Livermore, CA 94550, USA; 3Department of Molecular Medicine, Cleveland Clinic Lerner College of Medicine, Case Western Reserve University, Cleveland, OH 44195, USA; 4Case Comprehensive Cancer Center, Case Western Reserve University School of Medicine, Cleveland, OH 44106, USA

**Keywords:** PCSK9, LDLR, Molecular Dynamics, MM/PBSA, protein–protein interaction (PPI), PPI disruptive mutation, residue interaction network

## Abstract

Gain-of-function mutations in PCSK9 (proprotein convertase subtilisin/kexin type 9) lead to reduced uptake of LDL (low density lipoprotein) cholesterol and, therefore, increased plasma LDL levels. However, the mechanism by which these mutants reduce LDL reuptake is not fully understood. Here, we have used molecular dynamics simulations, MM/PBSA (Molecular Mechanics/Poisson–Boltzmann Surface Area) binding affinity calculations, and residue interaction networks, to investigate the protein–protein interaction (PPI) disruptive effects of two of PCSK9′s gain-of-function mutations, Ser127Arg and Asp374Tyr on the PCSK9 and LDL receptor complex. In addition to these PPI disruptive mutants, a third, non-interface mutation (Arg496Trp) is included as a positive control. Our results indicate that Ser127Arg and Asp374Tyr confer significantly improved binding affinity, as well as different binding modes, when compared to the wild-type. These PPI disruptive mutations lie between the EGF(A) (epidermal growth factor precursor homology domain A) of the LDL receptor and the catalytic domain of PCSK9 (Asp374Tyr) and between the prodomain of PCSK9 and the β-propeller of the LDL receptor (Ser127Arg). The interactions involved in these two interfaces result in an LDL receptor that is sterically inhibited from entering its closed conformation. This could potentially implicate the prodomain as a target for small molecule inhibitors.

## 1. Introduction

Cardiovascular disease is the number one killer in the United States; according to the Center for Disease Control and Prevention, nearly 650,000 people died of heart disease in 2017, accounting for roughly 1 in 4 deaths [1]. Autosomal dominant familial hypercholesterolemia (FH), defined by elevated LDL (low density lipoprotein) levels in which 70–95% of cases result from mutations in one of three genes (apolipoprotein B [APOB], low-density lipoprotein receptor [LDLR], proprotein convertase subtilisin/kexin type 9 [PCSK9]) [2], is associated with a substantial increase in coronary heart disease and atherosclerotic cardiovascular disease [3] and affects roughly 1 in 250 U.S. adults [4]. A recent meta-analysis of nearly 200,000 patients in statin therapy trials determined a 21% reduction in major vascular events for every 1.0 mmol/L reduction in LDL [5].

LDL has been shown to be the main driving force behind atherosclerotic cardiovascular disease [6,7]. Derived from VLDL (very low density lipoprotein) and IDL (intermediate density lipoprotein) particles, LDL carries the majority of circulating cholesterol [8]. Each LDL molecule has one apolipoprotein B as its only protein component [9], enveloping the surface of LDL and providing a macromolecular scaffold and structural integrity [7]. LDL clearance in the liver is mediated by the LDL receptor (LDLR). Briefly, the LDL receptor binds to Apo B on the surface of the LDL via its ligand binding domains. The LDLR/LDL complex is internalized into the cell via clathrin-mediated endocytosis. The LDL is released in endosomes due to the lowered pH, leading to eventual lysosomal degradation. LDLR enters a closed conformation and is recycled to the cell surface [10]. There are currently nearly 2000 reported variants in the LDLR gene [11], which are classified by the type of defect in the LDLR protein. These defects include no LDLR synthesis, no LDLR transport, no LDL cholesterol binding, no LDL cholesterol internalization, and no LDLR recycling [12]. These mutations are the most common cause of FH, followed by mutations to APOB.

The link between FH and the gene encoding proprotein convertase subtilisin/kexin type 9 (PCSK9), thereby establishing a third locus in FH, was first made in 2003 [13]. Patients in a French cohort with PCSK9 mutations displayed elevated LDL cholesterol levels. Shortly after, the binding site for PCSK9 to the EGF(A) (epidermal growth factor precursor homology domain A) domain of LDLR was determined [14], with evidence that the interaction rerouted LDLR to lysosomes for degradation as opposed to being recycled to the cell surface; crystal structures of the interaction [15,16] followed shortly thereafter. Within a few years, the crystal structures of the PCSK9/LDLR interaction including the entire EGFPH (epidermal growth factor precursor homology domain, consisting of EGF(A), EGF(B), the β-propeller, and EGF(C)) region of LDLR were solved [17], as well as a structure including nearly all of L7 of the ligand binding domain (PDB ID: 3M0C). A weak interaction between Leu108 of the prodomain of PCSK9 and Leu626 in the β-propeller of LDLR appears to stabilize the receptor in an open conformation, though the interface was described as making a marginal contribution to the interaction.

The initial study [13] identified two mutations, Ser127Arg and Phe216Leu, as disease-causing. Another mutation, Asp374Tyr, was discovered shortly thereafter [18,19]. The Ser127Arg and Asp374Tyr mutations have been shown to have a higher binding affinity for LDLR [20], while the Phe216Leu mutation has shown to be protective against deactivation by furin cleavage [21]. A fourth mutation, Arg496Trp, was found in another patient with elevated cholesterol [22]. A recent preprint suggests this mutation is unable to be bound by LDL cholesterol, thereby increasing the circulating PCSK9 available to bind LDLR [23].

The exact mechanism for the higher binding affinity in the Asp374Tyr and Ser127Arg mutants has yet to be elucidated. As these two mutations are implicated in a significant increase in circulating cholesterol, the mechanism by which these mutations increase the degradation of LDLR could be a target for therapeutic interventions for cardiovascular disease. Additionally, both are surface mutations on PCSK9 and are potentially implicated in perturbed protein–protein interactions. Here, we use molecular dynamics (MD) simulations, binding affinity calculations, and residue interaction network analysis to determine how these two gain-of-function (Figure 1a) mutations in PCSK9 influence the protein–protein interaction network effect with LDLR.

## 2. Results

### 2.1. RMSD/RMSF Analysis of PPI Disruptive Mutations on PCSK9 and LDLR Complex

Early in the MD simulation of the wild-type system of PCSK9 complexed to LDLR, the β-propeller moves away from the prodomain of PCSK9 because of the flexible nature of the LDL receptor’s EGF(A) and EGF(B) domains, causing an immediate spike in the overall root-mean-square deviation (RMSD). A slight contraction over the length of the EGF domains occurs shortly after, and the β-propeller comes in contact with a loop over residues 70–75 in the prodomain of PCSK9. Another spike in the RMSD occurs at roughly 260 nanoseconds; at this point in the simulation, the β-propeller rotates toward the C-terminal domain of PCSK9, rotating back shortly after and settling into its final conformation. The Asp374Tyr mutant engages the β-propeller very early in the simulation, affording a much lower overall RMSD, but otherwise has a similar interaction in this region.

The Ser127Arg mutation affords significantly higher stability early (as well as late) in the MD simulation. The interaction between the mutated residue and the β-propeller is immediate and present throughout the simulation. Between 130–165 nanoseconds of the simulation, there is a slight shift in this interaction, which then re-stabilizes and remains nearly constant throughout the remainder of the simulation. The Arg496Trp mutation similarly has a more marked deviation throughout the first ~325 nanoseconds of the simulation, at which point it settles into its final conformation. Early in the trajectory the β-propeller moves away from the prodomain of PCSK9, but the EGF(B) loop does not change conformations along with this movement, keeping this portion of the β-propeller presented to the prodomain. Interestingly, and in contrast to the wild-type and Asp374Tyr mutants, the region surrounding residue 127 interacts strongly with the β-propeller of LDLR in this mutant after moving back in contact with the prodomain.

The interface centered around the EGF(A) domain of LDLR remains consistent throughout all four simulations. An investigation of the RMSF of PCSK9 (Appendix A) shows higher fluctuations in this interface (centered around residue 374) in both the wild-type and Ser127Arg mutant when compared to the Asp374Tyr mutant, indicating a slight stabilization in this area due to the mutation. In both the wild-type and Asp374Tyr mutant, the L7 domain of LDLR (N-terminal in this structure) interacts with a loop between residues 165 and 180 on PCSK9, leading to a stabilization in these regions (Appendix A) when compared to the Ser127Arg mutant, for which this interaction does not occur. As previously reported, residues in the ligand binding domains of LDLR could potentially be important for PCSK9 binding [24]; however, due to the flexible nature of the ligand binding domains, our simulations cannot conclusively determine the significance of the interaction between L7 and PCSK9. Similarly, the loop between residues 210 and 220 have more flexibility in the wild-type than both mutants. Again, the simulations are inconclusive regarding differences in this loop because the loop is missing density in the crystal structure and is, therefore, modeled into the structure. 

Due to the fluctuations in RMSD early in the trajectories (as a result of the high flexibility of the EGF domains of LDLR), the root-mean-square fluctuation (RMSF) is also constructed for the final 200 nanoseconds of simulation (data not shown). The overall RMSF drops significantly for the wild-type over this portion of the trajectory, indicating the stabilization of the interaction also leads to a stabilization of the residues. Interestingly, the entire complex has less fluctuation in the wild-type over this portion of the trajectory than any of the mutants. The Asp374Tyr mutant displays marked flexibility in the EGF(B) domain, which is stabilized in the wild-type and Ser127Arg mutants due the loop folding back toward the β-propeller. While the C-terminus of LDLR, which would interface with a membrane surface, is stabilized by curling onto itself in the wild-type, Asp374Tyr, and Ser127Arg mutants, this region maintains an extended conformation in the Arg496Trp mutant, resulting in higher fluctuations. Overall, no significant changes in the RMSF are noted as being concomitant with each simulation attaining a stable conformation, and the overall decrease in RMSF from residues in loops is expected for such a stable structure.

### 2.2. Ser127Arg/Asp374Tyr Improve the PCSK9-LDLR Interaction by MM/PBSA Analysis

The final 50 ns of each simulation were used for each analysis. Each system was sampled every 500 ps, resulting in 101 frames being used. The resulting binding affinities qualitatively match binding data from previous in vitro studies with immobilized LDLR [20] (Table 1), which indicate an increase in binding affinity for both Ser127Arg and Asp374Tyr mutants over wild-type. Furthermore, the affinity for the Asp374Tyr mutant is higher than the Ser127Arg mutant, matching the previous assay. As a positive control, the Arg496Trp mutant was included in the MM/PBSA calculations. We did not anticipate any conformational change to occur related to this mutation; however, it is a clinically relevant mutation which would allow us to further understand the impact changing the charge distribution (replacing a positive residue with a neutral residue) could have on the binding affinity. While the difference in the binding affinity between the wild-type and the Arg496Trp mutant is not as great as for the PPI disruptive mutations, the binding affinity still increases over the wild-type. To determine why, we decomposed the affinity by interaction type, domain, and by residue. It should be noted that a per-residue decomposition of the polar and non-polar solvation energies will not add up to the respective total energies, but the differences do not impact the qualitative nature of the MM/PBSA calculations.

While the contribution to the binding affinity from the molecular mechanics energy decreases in the Ser127Arg mutant versus wild-type (van der Waals plus electrostatic), the polar solvation energy contribution to the binding increases (smaller positive number = better binding). In the Asp374Tyr mutant, the polar and non-polar solvation energies contribute slightly less to the binding affinity than the wild-type, but the total molecular mechanics energy contribution increases significantly. Interestingly, the total molecular mechanics binding was lowest for the Arg496Trp mutant, while also having the most favorable value for polar solvation.

### 2.3. A direct Contribution by Ser127Arg and Asp374Tyr to the PCSK9-LDLR Interaction by the Domain Decomposition of MM/PBSA

As a second qualitative measure, the binding affinity was separated for the domains of each protein to further identify how the affinity changes due to mutation (Table 2, Appendix A). Furthermore, the interactions can be combined as PCSK9 prodomain with the LDLR β-propeller and PCSK9 catalytic/C-terminal domain (catalytic) with the EGF(A), EGF(B), and L7 domains (loop) of LDLR. A comparison of the prodomain/β-propeller interactions reveals the wild-type and Asp374Tyr mutant have a similar affinity in this region, while the affinity in the Arg496Trp mutant is improved over wild-type, and the Ser127Arg mutant’s interaction in this region is even further enhanced. Similarly, the interaction between the loop and catalytic domain in the Asp374Tyr mutant is much higher than in the wild-type, and even further enhanced when compared to the Ser127Arg mutant. The Arg496Trp mutant is the least favorable interactor in this region. The difference between the Ser127Arg mutant and wild-type can be attributed to the L7 domain of the wild-type system interacting with PCSK9, while in the Ser127Arg mutant this terminal loop remains away from PCSK9, as in the crystal structure. This interaction is present in the Asp374Tyr mutant as well; however, this does not explain the large discrepancy with the Arg496Trp mutant.

Looking into the impact the difference in the total charge of the systems has on the binding lends some clarity to some of the values. For instance, the C-terminal domain of PCSK9 (residues 450–682 in this crystal structure) do not have any interaction with the LDL receptor. Because of this, it would be expected that these residues would have little to no impact on the binding affinity. However, there is a contribution made due to long-range electrostatic interactions and the lack of a cut-off to the vacuum molecular mechanics interaction energy in MM/PBSA calculations. The Asp374Tyr and Ser127Arg mutants, which have the lowest overall system charge, have the highest contribution to the binding affinity (−91 and −95 kJ/mol, respectively). The wild-type system has one fewer positively charged residue than each of these mutants with a contribution of −78 kJ/mol from the C-terminal domain. The Arg496Trp mutation removes a positively charged arginine in the C-terminal domain, replacing it with a neutral tryptophan. This results in a drop in the contribution from the C-terminal domain to −37 kJ/mol. To further our understanding of the mechanism behind the PPI disruptive interactions, we turned to residue interaction networks.

### 2.4. Residue Interaction Network Analysis Indicates Improved PCSK9/LDLR Binding by Ser127Arg and Asp374Tyr

Residue interaction networks (RINs) were implemented in an effort to determine how these point mutations impacted the local interaction networks between PCSK9 and LDLR. The networks used here are generated from 251 frames over the final 50 ns of simulation (every 200 picoseconds). Residues are considered to be in contact with each other if the sum of the van der Waals radii of each atom minus the distance between the atoms is greater than −0.4 angstroms (Å):overlap_ij_ = r_VDWi_ + r_VDWj_ − d_ij_(1)

The residue interaction network between each set of proteins paints a clear picture as to the nature of the differential interaction. In the crystal structure, only one residue (LEU108) in the prodomain of PCSK9 is within four angstroms of any part of the LDL receptor. Over the final 50 ns of simulation, the wild-type PCSK9 has slightly more interaction than the Asp374Tyr substitution in the prodomain, though neither has any interaction surrounding residue 108 (Figure 2a). The degree differences in the prodomain/ β-propeller interaction can be attributed to Gln453 in the β-propeller of the LDL receptor. While this residue has four edges (interactions) in the wild-type, it only has one in the Asp374Tyr mutant. While care should be taken when comparing the impact of singular residues on the binding affinity using MM/PBSA, the contribution from Gln453 is higher in the wild-type than in the Asp374Tyr mutant (−10.31 kJ/mol vs. 0.79 kJ/mol, respectively). 

The mutation of Ser127, however, leads to a different conformation for the prodomain/β-propeller interaction, thereby revealing a significantly different RIN (Figure 2b). While Pro71 in the prodomain of PCSK9 makes a number of contacts with LDLR in the wild-type and Asp374Tyr mutant, the interaction is not present in the Ser127Arg mutant. The MM/PBSA energies reflect this with a value of 0.11 kJ/mol for the Ser127Arg mutant −14.16 kJ/mol for wild-type. Instead, we see a number of interactions centered around the mutated residue, whereas the wild-type (Figure 2c) and Asp374Tyr (not shown) simulations without mutations at Ser127 have no interactions at all. While the Ser127 residues of the wild-type and Asp374Tyr mutant do make a small contribution (−1.28 kJ/mol and -2.02 kJ/mol, respectively) to the binding affinity due to the Lennard-Jones and Coulombic interactions, the mutation to arginine leads to a significant, direct contribution (−141.75 kJ/mol) based on binding affinity and number of interacting partners. 

Residue 108 has very important interactions in the Arg496Trp mutant (Figure 2d) although the interactions surrounding residue 108 are not present in the wild-type and Asp374Tyr mutants. While the residues in the prodomain contributing most to the PCSK9/LDLR interaction in the wild-type and Asp374Tyr mutant have zero contact in Arg496Trp, Leu108 makes three contacts. Interestingly, Ser127 makes four contacts with LDLR; this affirms that even when the interaction surrounding residue 127 is maintained, the mutation to arginine directly impacts the strength of interaction; the contribution to the binding affinity in Ser127 is on par with the wild-type and Asp374Tyr substitution (−1.44 kJ/mol). The region directly surrounding residue 127 in the wild-type (Figure 3a) has nearly no interaction, while in the Arg496Trp (Figure 3b) and Ser127Arg (Figure 3c) mutants this region interacts with LDLR. 

Although the difference in the networks is less pronounced in the Asp374Tyr mutant compared to the Ser127Arg mutant, there is still clear evidence for the importance of the mutant to the interaction (Figure 4a). Both the wild-type (Figure 4b) and Ser127Arg (not shown) mutant lose the interactions present in the crystal structure at residue 374, while the Arg496Trp mutant maintains a single contact (not shown); however, the mutation to Tyr allows for three interactions with the LDL receptor (Figure 4c). Most importantly, a tight (~1.94 Å) hydrogen bond is between the phenyl oxygen on Tyr374 and the imidazole ring of the nearby His306 on the LDL receptor, whereas the wild-type structure has no interaction partners (Figure 5a,b). The mutation causes a slight shift in the RIN, where Leu318 in LDLR no longer interacts with Cys378 in PCSK9 (instead interacting with Tyr374), though the remainder of the RINs are similar.

### 2.5. Alternative Conformations for the PCSK9/LDLR Interaction 

In replicate simulations, we were able to find a second conformation for the PCSK9/LDLR interaction which did not involve the prodomain/β-propeller interaction. This second conformation results in a lower overall affinity between PCSK9 and LDLR (data not shown) compared to the simulations including the prodomain/β-propeller interaction. While in the Arg496Trp mutant simulation the LDLR remains in an extended conformation (Appendix A), a replicate of the Asp374Tyr mutant results in the β-propeller curling back into the EGF domains (Appendix A), which would be an unlikely result in the presence of a membrane surface. However, this does provide some evidence that it is possible for PCSK9 to bind LDLR without preventing a closed conformation. A mapping of the crystal structure of LDLR in its closed conformation (PDB ID: 1N7D [10]) indicates that the interaction between the β-propeller and the ligand binding domains is not sterically inhibited in this conformation, though a rotation in EGF(B) would be required. 

## 3. Discussion

Here we have proposed a plausible mechanism behind the disease-causing PPI disruptive mutations Ser127Arg and Asp374Tyr on PCSK9 and LDLR complex. Both of these mutations directly impact the protein–protein interaction upon which they lie in the number/type of interactions made and the concomitant improvement to binding affinity in the region. The interaction in the Ser127Arg mutation could have clinical relevance; as this interaction (prodomain of PCSK9 and β-propeller of LDLR) appears to be the driving force behind preventing LDLR from entering a closed conformation [10], it seems there could be potential for targeting this region with therapeutics inhibitory of the protein–protein interaction here. 

Although the Asp374Tyr mutant does not directly improve the binding affinity with the β-propeller of the LDL receptor, it is clear that the mutation improves the interaction with the EGF(A) domain of LDLR. This significant improvement to the binding affinity would lead to more PCSK9-LDLR interacting partners, increasing the chances of a secondary interaction in the β-propeller occurring. Conversely, while the Ser127Arg mutation does not improve the interaction with EGF(A), there is a clear increase in affinity between the prodomain of PCSK9 (where the mutant lies) and the β-propeller, improving the chance of preventing the receptor from entering a closed conformation.

Caution must be exercised when interpreting the results of MM/PBSA for such large, electrostatically heterogeneous systems. Charged residues (arginine, lysine, aspartic acid, glutamic acid) will have a disproportionate impact on the binding affinity, regardless of if the residue is on the protein–protein interface or nanometers away. We expected the overall binding affinity calculations to result in positive values based on the overall charge of each protein, and the method for calculating the interaction energy, with PCSK9 carrying a −3 charge (−2 in Ser127Arg and Asp374Tyr mutants and −4 in Arg496Trp mutant) and LDLR having −14 charge. Further, decomposing the affinity inherent in PCSK9 based on domain (prodomain vs. catalytic/C-terminal) would result in a large negative value for the prodomain (+3 in Ser127Arg mutant, +2 in all others) and large positive value for the remainder of the protein, as presented here. While a more quantitative value for binding affinity would surely be preferred, the computational barrier to such a calculation is quite high, without necessarily leading to more accurate results. 

As noted above, we must also take care when evaluating contributions to the binding affinity on a per-residue basis. For a typical protein-ligand complex, especially for an uncharged ligand, it is expected that all non-interface residues will have an interaction energy near zero. However, the introduction of charged particles, and in our case numerous charged residues can cause a significant obfuscation of results. The high number of negatively charged residues at biological pH in the LDL receptor causes all arginine and lysine residues to have a much higher binding affinity than would be expected, even for residues not on the interface. This is coupled with a large positive value for all aspartic and glutamic acid residues. As examples, Arg659 and Asp660 in the PCSK9 C-terminal domain, which has no interaction with LDLR, both have near zero contributions due to polar and non-polar interactions. However, their molecular mechanics (van Der Waals and electrostatics combined) contributions are −64.3 kJ/mol and 72.7 kJ/mol, respectively. Conversely, while those residues in LDLR closest to the catalytic domain (EGF(B), EGF(A), L7) experience the same extreme values, residues in the β-propeller have an inversion of sign in the residues mentioned above; this is due to the overall slight positive charge on the prodomain. Despite these challenges, we have demonstrated the impact these PPI disruptive mutations (Ser127Arg and Asp374Tyr) have on the interaction between PCSK9 and LDLR.

While it is clear that these mutations directly impact the binding affinity in these regions, it has yet to be demonstrated if both interactions (catalytic domain of PCSK9/EFG(A) of LDLR and prodomain/β-propeller) are necessary to induce the degradation of the LDL receptor. Through our replicated simulations, we found that conformations where the prodomain/β-propeller interaction did not exist were possible; however, simulation studies are not enough to determine if preventing this interaction is sufficient to prevent LDLR degradation by PCSK9. Further functional studies would be required to determine if interference of this interaction could potentially rescue the phenotype, especially in those with the Ser127Arg mutation.

## 4. Materials and Methods 

### 4.1. Selection of Disease-Causing Mutations

To determine a set of disease-causing mutations for study, a literature search was carried out. From our search, the Ser127Arg and Asp374Tyr mutants were found to be in numerous cohorts, and also fit our requirement of PPI disruptive mutations. A third mutation was chosen to be a positive control; at the time of running the simulations, the mechanism for the increased LDLR degradation from the Arg496Trp mutant had yet to be elucidated. Since the “gain-of-function” mechanism for the Phe216Leu mutation was known (reduced furin cleavage), we chose the Arg496Trp mutant.

### 4.2. System Construction

The crystal structure (PDB: 3M0C) was accessed from the RCSB PDB protein data bank [25]. This structure was chosen in an attempt to not bias the results for the Ser127Arg mutant; the distance between the prodomain of PCSK9 and the β-propeller of LDLR is slightly higher in this structure when compared to other co-crystallized structures available from RCSB. Co-crystalized calcium ions were retained from the structure, and were not involved in the protein–protein interaction. Non-terminal missing loops were reconstructed using Modeller9.18 [26] within UCSF Chimera [27]. Protonation states for charged residues were determined at neutral pH using PROPKA 2.0 [28]. Mutations and preparation of the system for molecular dynamics simulation were accomplished using the quick MD simulator module of CHARMM-GUI [29,30]. Following a processing step, including adding hydrogens and patching the terminal regions, a water box using TIP3P water molecules with edges at least 12 Å from the protein was added. The system was neutralized to an NaCl concentration of 150 mM. Each system was composed of roughly 350,000 atoms. The Arg496Trp and Asp374Tyr systems were both simulated twice while all other systems were simulated once; the initial simulations did not result in an interaction between the β-propeller of LDLR and the A domain of PCSK9, and so were repeated to obtain more comparable results.

### 4.3. Simulation Parameters

MD simulations were carried out using GROMACS 2018.2 [31] with the CHARMM36m force field [32] on the Pitzer computing cluster at the Ohio Supercomputer Center. Initial minimizations of the systems were carried out using steepest descent until the energy of the system reached machine precision. Following minimization, an NVT equilibration step with positional restraints of 400 kJ mol^−1^ nm^−2^ on backbone atoms and 40 kJ mol^−1^ nm^−2^ on side chain atoms was run using a timestep of 2 fs for 500,000 steps, yielding 1 ns of equilibration. Finally, NPT dynamics were run with no positional restraints for 500 ns using the same 2 fs timestep from equilibration. 

Hydrogen atoms were constrained using the LINCS [33] algorithm. Temperature coupling to 310.15 K was done separately for the protein and the water/ions using a Nose-Hoover thermostat [34] and a 1 ps coupling constant. For the NPT dynamics simulation, isotropic pressure coupling to 1 bar was done using a Parrinello-Rahman barostat [35] with a coupling constant of 5.0 ps and compressibility of 4.5e-05 bar^−1^. The pair-list cutoff was constructed using the Verlet scheme [36], updated every 20 evaluations with a cutoff distance of 1.2 nm. Particle mesh Ewald [37] electrostatics were chosen to describe coulombic interactions using the same cutoff as in the pair-list. van der Waals forces were smoothly switched to zero between 1.0 and 1.2 nm using a force-switch modifier to the cut-off scheme. 

### 4.4. System Analysis

All four systems (wild-type, Ser127Arg, Asp374Tyr, Arg496Trp) were minimized and equilibrated as described in the methods section, then submitted to the Ohio Supercomputer Center for simulation. Simulations were run in 5 separate 100 ns increments, totaling 500 nanoseconds for each system. Standard GROMACS tools were used for all post-processing, including concatenation of the trajectories and re-centering of the system in the periodic box. Analyses of the RMSD (Figure 1b) and RMSF (Appendix A) were carried out using GROMACS tools. RMSD was calculated over all backbone atoms after least-squares fitting to the same, while the RMSF was calculated per residue and separated by protein after least-squares fitting to backbone atoms.

MM/PBSA energies were calculated on 101 frames over the final 50 ns of each simulation using g_mmpbsa 1.6 [38], which uses APBS 1.3 [39] to determine the polar and non-polar contributions to the binding energy. Briefly, the binding free energy can be expressed as
(2)$$\Delta G_{binding}=G_{complex}-\left(G_{protein1}+G_{protein2}\right)$$
where complex refers to the protein–protein complex, and protein 1 and protein 2 the respective proteins in the complex. The individual free energies for each component above are determined by
(3)$$G_{x}=E_{MM}+G_{solvation}-TS$$
where $E_{MM}$ is the vacuum molecular mechanics energy, $G_{solvation}$ the solvation energy, and TS the entropic contribution. Entropic contributions were not included owing to computational cost and evidence that the inclusion of the entropy term does not always improve the accuracy of the calculations [40]. The molecular mechanics energy and solvation energy can be further broken down into their component energies:(4)$$E_{MM}=E_{bonded}+E_{nonbonded}=E_{bonded}+E_{vdW}+E_{elec}$$
(5)$$G_{solvation}=G_{polar}+G_{nonpolar}$$

Here, $E_{bonded}$ is zero, since we have used the single trajectory approach. $E_{vdW}$ and $E_{elec}$ are the van der Waals and electrostatic contributions to the vacuum binding, respectively, while $G_{polar}$ and $G_{nonpolar}$ are the electrostatic and non-electrostatic contributions to the solvation energy. The solute dielectric constant was set to 2 for all systems.

Graphics were generated using Chimera 1.13.1 and data plots in ggplot2. Residue interaction networks were generated using the StructureViz2 [41] addon within Cytoscape 3.7 [42], which allows us to visualize the results generated by Chimera (Appendix A). 

## Figures and Tables

**Figure 1 ijms-21-01550-f001:**
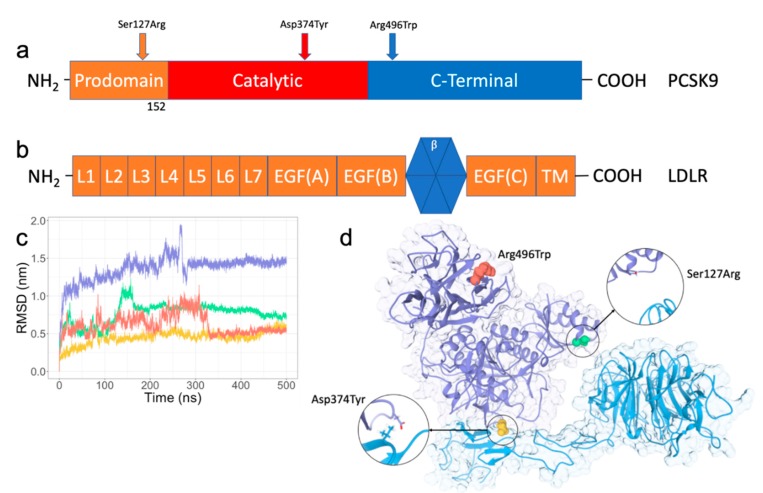
Depiction of the studied mutants. (**a**) Domain composition of proprotein convertase subtilisin/kexin type 9 (PCSK9). The prodomain (residues 31-152) is autocleaved and remains non-covalently bound to the catalytic domain. (**b**) Domain composition of low-density lipoprotein receptor (LDLR). The ligand binding domain is composed of seven (L1-L7) repeats, each roughly 40 amino acids in length. The epidermal growth factor precursor homology domain (EGFPH) is composed of epidermal growth factor precursor homology domain A (EGF(A)), the β-propeller, EGF(B), and EGF(C), and followed by a transmembrane region. (**c**) Root-mean-squared deviation (RMSD) plot for each trajectory over each 500-nanosecond production run. Wild-type is in purple, Ser127Arg is green, Asp374Tyr is yellow, and Arg496Trp is red. (**d**) Crystal structure of the PCSK9/LDLR complex (PDB ID: 3M0C). Clinically relevant mutations investigated in this study are depicted as spheres. Insets indicate the protein–protein interactions incident at positions 127 and 374.

**Figure 2 ijms-21-01550-f002:**
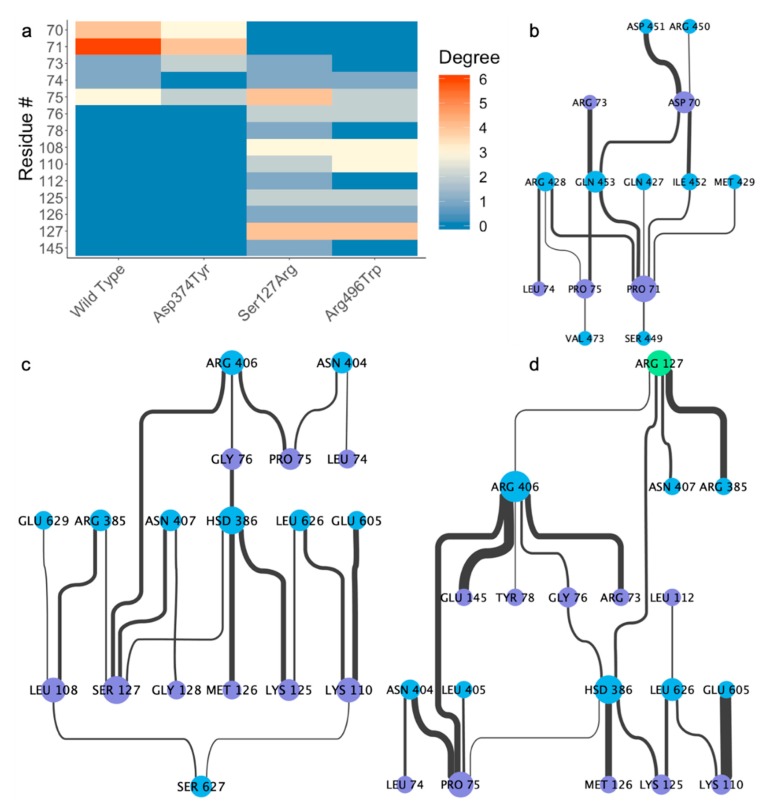
Contacting residues in the prodomain of PCSK9. (**a**) Number of contacts. (degree) in the prodomain of PCSK9 in each system. No residues in the crystal structure met the contact criteria used. Residue interaction network between the prodomain of PCSK9 and LDLR in the (**b**) wild-type, (**c**) Arg496Trp, and (**d**) Ser127Arg systems. Edge weight indicates strength of interaction, and node size indicates number. of interactions (connectivity or degree in residue-residue networks).

**Figure 3 ijms-21-01550-f003:**
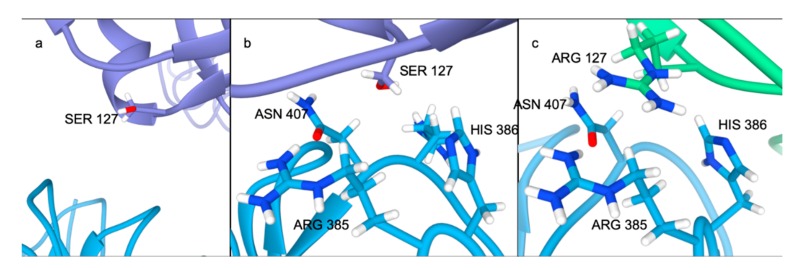
Interface surrounding residue 127 in the (**a**) wild-type, (**b**) Arg496Trp, and (**c**) Ser127Arg systems at the end of 500 nanoseconds of simulation.

**Figure 4 ijms-21-01550-f004:**
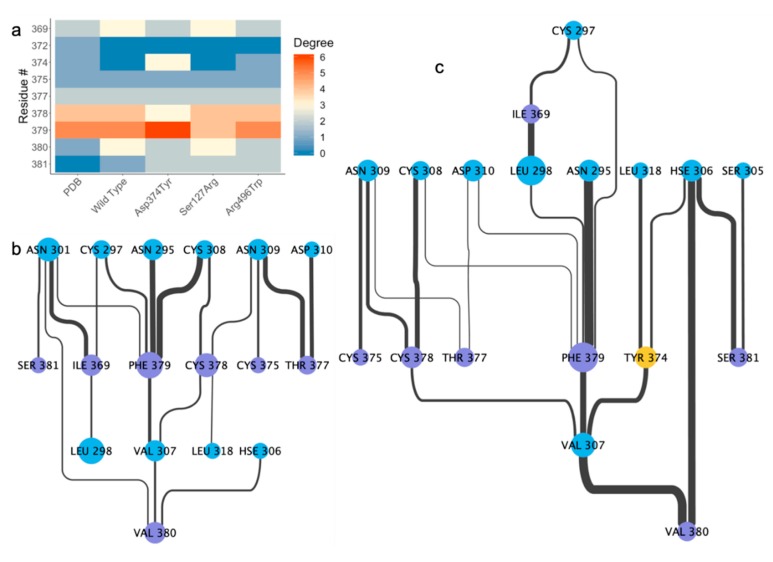
Contacting residues in the catalytic domain of PCSK9. (**a**) Number of contacts (degree) of residues in PCSK9 along the crystallized interface of the catalytic domain of PCSK9 and EGF(A) in LDLR. Residue interaction network for the interface between the catalytic domain of PCSK9 and LDLR in the (**b**) wild-type and (**c**) Asp374Tyr systems.

**Figure 5 ijms-21-01550-f005:**
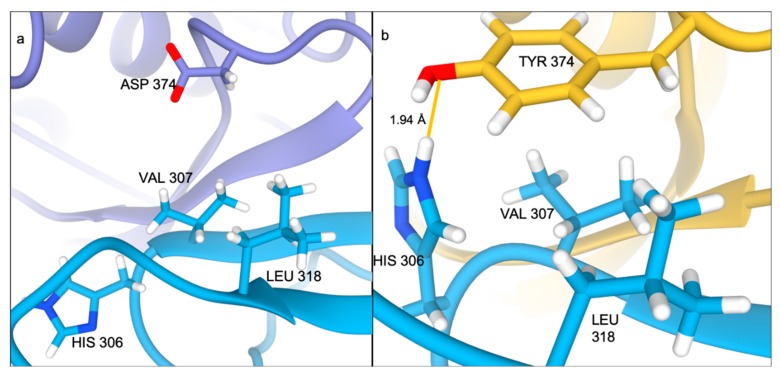
Interface of the (**a**) wild-type and (**b**) Asp374Tyr mutant at the PCSK9 catalytic domain/LDLR interface after 500ns of MD simulation.

**Table 1 ijms-21-01550-t001:** Decomposition of the MM/PBSA energies by interaction type (uncertainty in parentheses).

(kJ/mol)	Wild-type	Asp374Tyr	Arg496Trp	Ser127Arg
van der Waals	−738.7 (3.78)	−544.8 (4.62)	−495.0 (2.97)	−495.8 (3.49)
Electrostatic	−227.6 (18.5)	-851.6 (23.3)	133.5 (16.5)	−391.1 (15.4)
Polar Solvation	1562.6 (24.1)	1615.5 (26.9)	878.4 (18.6)	1138.7 (17.3)
Non-polar	−89.7 (0.593)	−76.8 (0.723)	−64.0 (0.507)	−64.2 (0.525)
Total	506.5 (18.7)	143.0 (12.0)	453.3 (13.5)	187.4 (14.2)

**Table 2 ijms-21-01550-t002:** A per-residue decomposition of the binding affinities grouped by domain and interaction.

(kJ/mol)	Wild-type	Asp374Tyr	Arg496Trp	Ser127Arg
PCSK9 Prodomain	−147.5	−124.3	−281.9	−417.2
PCSK9 Catalytic	375.7	185.9	506.4	438.1
Loop	248.9	93.9	290.6	250.6
LDLR β-propeller	4.6	−41.2	−77.9	−106.4
Prodomain/β-propeller	−143.0	−165.5	−359.8	−523.6
Catalytic/Loop	624.6	279.8	797.0	688.7

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
