# Peer review of "In Silico Insights into Protein–Protein Interaction Disruptive Mutations in the PCSK9-LDLR Complex"

_ijms, 2020, doi:10.3390/ijms21051550_

Round 1

Reviewer 1 Report

Ms ijms-706922:

PCSK9 has been revealed as a promising alternative in the regulation of cholesterol levels in blood. Thus, PCSK9 binds to LDL receptor and targets it for lysosomal degradation. In the present study, the contribution of two previously described gain-of-function mutations in PCSK9 (S127R and D374Y)  to the binding affinity to LDLR is reported, based on MD simulations and MM/PBSA approach. In both cases the binding affinity is increased compared to the wild type, in agreement with previous in vitro studies. In addition, a detailed energetic dissection at the residue level is carried out in order to understand the basis for the molecular recognition of this relevant complex.

In my opinion some issues need further clarification:

- A significant part of the discussion is devoted to the impact of electrostatic interactions in the binding affinity. This part is confusing not knowing the specific charged residues and, furthermore, their location in the structure of both, LDLR and PCSK9. It is said that both molecules have a negative net charge. Nevertheless, in order to get some insights into the role of electrostatics in binding It would be useful to have a look at the specific charge-charge interactions and their location.  

- No notice has been made regarding the pH of the calculations. I assume is neutral.

It is well known that pH driven conformational change of the LDLR is relevant in vivo. Have the authors tried to get an estimate of the pH effect on their calculations??

- In line 25 : “combination of these TWO INTERACTIONS results in an LDL receptor that is sterically inhibited from entering its closed conformation” . It would be more appropriate the term “interactions involved in these two interfaces” of contact between LDLR and PCSK9 (the beta-propeller/prodomain interface and EGF (A)/catalytic domain interface).

- It would be very useful if the domain composition of LDLR and PCSK9 in Fig 1d could be distinguished, maybe with color code.

- Table 1: In my opinion the contribution of R496W (the control) to the total DG binding is, within the error, the same as the wild type (453.262±13.5 versus 506.528±18.7 kJ/mol). Trying to “further understand the impact of changing the charge distribution (removing just one positive charge!) on the binding affinity (lines 151-153)” based on this small difference is going beyond the data. In this case, it is hard to believe that just a single mutation located far away from the interface with LDLR could make a difference (although I am aware cases of long-distance communication have been reported) in the binding.

Also, the magnitudes and their errors should have the same number of decimal figures.

- Line 65: “small” should be replaced by “weak”.

- Line 194: “to further INCREASE our understanding…”, verb is missed.

- Line 205: Leu 108 instead of LEU 108.

- Line 277: “LDLR” instead of LDLD complex.

- Line 276: “Here we have demonstrated the mechanism behind the disease-causing PPI disruptive mutations…” should be replaced by “here we proposed a plausible mechanism…”. Experimental work should be performed in order to demonstrate their theoretical calculations.

Author Response

A significant part of the discussion is devoted to the impact of electrostatic interactions in the binding affinity. This part is confusing not knowing the specific charged residues and, furthermore, their location in the structure of both, LDLR and PCSK9. It is said that both molecules have a negative net charge. Nevertheless, in order to get some insights into the role of electrostatics in binding It would be useful to have a look at the specific charge-charge interactions and their location.

The intention within this portion of the discussion was to explain the positive values for binding affinity stemming from the MM/PBSA calculations.  Of course, in reality the values should be of much smaller magnitude and negative.  An example of a non-interface residue with high calculated binding affinity has been added.

No notice has been made regarding the pH of the calculations. I assume is neutral.

This is correct.  The pH has been noted in the methods.

It is well known that pH driven conformational change of the LDLR is relevant in vivo. Have the authors tried to get an estimate of the pH effect on their calculations??

We had considered this, and had constructed a system at a pH of 6.  However, the lowered pH led to a system with a much lower net charge.  Due to the nature of the electrostatic contribution to the MM/PBSA calculation, a comparison between pH states would have been significantly obfuscated by the charge differential in non-interface residues.

In line 25 : “combination of these TWO INTERACTIONS results in an LDL receptor that is sterically inhibited from entering its closed conformation” . It would be more appropriate the term “interactions involved in these two interfaces” of contact between LDLR and PCSK9 (the beta-propeller/prodomain interface and EGF (A)/catalytic domain interface).

Thank you for the suggestion.  This has been adjusted.

It would be very useful if the domain composition of LDLR and PCSK9 in Fig 1d could be distinguished, maybe with color code.

Our concern is that adding more colors here could make the image more confusing due to the coloration in the mutated residues.  We do believe it would be reasonable to add as a supplementary figure (sans mutation sites), referenced at the beginning of section 2.2.

Table 1: In my opinion the contribution of R496W (the control) to the total DG binding is, within the error, the same as the wild type (453.262±13.5 versus 506.528±18.7 kJ/mol). Trying to “further understand the impact of changing the charge distribution (removing just one positive charge!) on the binding affinity (lines 151-153)” based on this small difference is going beyond the data. In this case, it is hard to believe that just a single mutation located far away from the interface with LDLR could make a difference (although I am aware cases of long-distance communication have been reported) in the binding.

I believe we agree on this point; however, due to the mechanism by which MM/PBSA is calculated, there is a decrease in binding affinity conferred by the Arg496Trp mutation of nearly 64 kJ/mol (-64.4 kJ/mol in wild type and -0.6 in the mutated system).  Of course, with no actual interaction present, there should be no change in binding affinity due to this residue in reality.

 Also, the magnitudes and their errors should have the same number of decimal figures.

This has been adjusted.

Line 65: “small” should be replaced by “weak”.

You are correct.  Fixed.

Line 194: “to further INCREASE our understanding…”, verb is missed.

Further is the verb in this sentence.

Line 205: Leu 108 instead of LEU 108.

Thank you for catching this.

Line 277: “LDLR” instead of LDLD complex.

And this.

Line 276: “Here we have demonstrated the mechanism behind the disease-causing PPI disruptive mutations…” should be replaced by “here we proposed a plausible mechanism…”. Experimental work should be performed in order to demonstrate their theoretical calculations.

You are correct; the previous wording was too strong and would require further validation.

Reviewer 2 Report

In this paper, the researchers conducted a series of in silico methods to investigate the protein-protein interaction disruptive effects of two of LCSK9's gain-of-function mutations, which can lead to reduced uptake of LDL. Overall the paper was well organized with well-designed experiments, comprehensive data and analysis, making it ready for publication.

Author Response

We thank the reviewer for their kind words, and for taking the time to review.